# Design, development and optimization of sustained release floating, bioadhesive and swellable matrix tablet of ranitidine hydrochloride

**Birhanu Nigusse[1,2], Tsige Gebre-Mariam[1,2]\*, Anteneh Belete[1,2]**

**1** Department of Pharmaceutics and Social Pharmacy, School of Pharmacy, College of Health Sciences, Addis Ababa University, Addis Ababa, Ethiopia, **2** Regional Bioequivalence Center, School of Pharmacy, College of Health Sciences, Addis Ababa University, Addis Ababa, Ethiopia

\* tsige.gmariam@aau.edu.et

**Data Availability Statement:** All relevant data are within the manuscript and its Supporting Information files.

## Abstract

Ranitidine HCl, a selective, competitive histamine $H_2$-receptor antagonist with a short biological half-life, low bioavailability and narrow absorption window, is an ideal candidate for gastro-retentive drug delivery system (GRDDS). Controlled release with an optimum retentive formulation in the upper stomach would be an ideal formulation for this drug. The aim of the present study was therefore to develop, formulate and optimize floating, bioadhesive, and swellable matrix tablets of ranitidine HCl. The matrix tablets were prepared using a combination of hydroxypropyl methylcellulose (HPMC) and sodium carboxymethyl cellulose (NaCMC) as release retarding polymers, sodium bicarbonate ($NaHCO_3$) as gas generating agent and microcrystalline cellulose (MCC) as direct compression diluent. Central composite design (CCD) was used to optimize the formulation and a total of thirteen formulations were prepared. Concentration of HPMC/NaCMC (3:1) ($X_1$) and $NaHCO_3$ ($X_2$) were selected as independent variables; and floating lag time ($Y_1$), bioadhesive strength ($Y_2$), swelling index at 12 h ($Y_3$), cumulative drug release at 1 h ($Y_4$), time to 50% drug release ($t_{50\%}$) ($Y_5$) and cumulative drug release at 12 h ($Y_6$) were taken as the response variables. The optimized batch showed floating lag time of 5.09 sec, bioadhesive strength of 29.69 g, swelling index of 315.04% at 12 h, $t_{50\%}$ of 3.86 h and drug release of 24.21% and 93.65% at 1h and 12 h, respectively, with anomalous release mechanism. The results indicate that sustained release matrix tablet of ranitidine HCl with combined floating, bioadhesive and swelling gastro-retentive properties can be considered as a strategy to overcome the low bioavailability and *in vivo* variation associated with the conventional ranitidine HCl tablet.

## Introduction

The oral route of drug administration is the most effective and widely used method of drug delivery because of its high patient compliance, cost-effectiveness, flexibility in the design of dosage form and ease of production [1]. However, this delivery approach has some physiological limitations including: a limited gastrointestinal transit time, an unpredictable level of

**Funding:** BN would like to thank the Regional Bioequivalence Center for sponsoring his MSc study, and School of Pharmacy, College of Health Sciences, Addis Ababa University for the financial and material support. BN would also like to thank Cadila Pharmaceutical Factory PLC for material support.

**Competing interests:** The authors have declared that no competing interests exist.

gastric emptying which varies from person to person, and the presence of an absorption window for several drugs in the upper part of the gastrointestinal tract (GIT). Such difficulties prompted researchers to develop a delivery system called the gastro-retentive drug delivery system (GRDDS) which allows the medicament to stay in the stomach for a prolonged and predictable period of time [2]. The GRDDS has many advantages such as reduced fluctuations in plasma drug levels, increased gastric retention times, enhanced drug absorption within the stomach, higher bioavailability, an improvement in therapeutic effectiveness, controlled release of active substances in the upper part of gastrointestinal (GI) tract and improvement of reduction in drug administration frequency [3–5]. Drug candidates that can benefit from such approach include (a) drugs with narrow absorption window in the upper part of the GIT, (b) acidic drugs, (c) drugs for local action in the stomach, (d) drugs that degrade at higher pH, (e) drugs with poor solubility at higher pH, (f) drugs which degrade in the intestine and colon, and (g) drugs with low biological half-life [6].

Different GRDDS approaches, namely a) co-administration of the drug delivery system with pharmacological agents that slow gastric motility, b) bioadhesive systems, c) size increasing systems which are either due to expansion or swelling and shape modification, d) density-controlled systems which are either, high density systems or floating systems, and e) magnetic systems have been reported [7]. Floating, bioadhesive and swelling/expandable systems are the most commonly used approaches among the retention mechanisms and extensive research have been conducted on these approaches. However, each of the individual retention approaches have their own limitations. Floating systems are highly dependent on the presence of food and gastric contents (they need high levels of gastric fluid in the stomach); efficiency of bioadhesive systems can be reduced by constant turnover of the mucus; and expandable systems are associated with problems related to matrix integrity (difficult to hold mechanical shape) [7]. This indicates that the use of a single GRDD approach may not overcome the low bioavailability of the drug, rapid gastric emptying and associated *in vivo* variations. According to recent studies, combining different gastroretentive mechanisms is an effective approach to increase gastroretentive capabilities and may overcome the limitation associated with individual retention mechanism [8, 9]. Hence, the delivery of candidate drugs using the combination of the 3 most commonly used retention mechanisms, namely floating, bioadhesive and swelling can be considered as a formulation strategy in GRDDS. Polymers with swelling and bioadhesive properties such as hydroxypropyl methylcellulose (HPMC), Carbopol, sodium carboxymethyl cellulose (NaCMC), hydroxypropyl cellulose (HPC) and effervescent agents like sodium bicarbonate can be used for this purpose, and a formulation with desired floating, bioadhesive and swelling and release properties can be prepared.

Ranitidine HCl, a selective, competitive histamine $H_2$-receptor antagonist, which is commonly employed in the management and treatment of many diseases in the upper GIT is among the ideal drugs for GRDDS. It has short biological half-life (~2.5–3 h) and low bioavailability (50–60%) [10]. It has also site-specific absorption from the upper part of the GIT and its bioavailability is significantly lower when administered to the colon due to colonic metabolism [11–13]. Moreover, it is frequently administrated in certain disease conditions (e.g., in erosive esophagitis). Hence, the drug has been suggested as a candidate for GRDDS. Subsequently, a few attempts have been made to develop the dosage form of ranitidine HCl using individual gastro-retentive mechanisms, e.g., floating tablets [14] and bioadhesive tablets [15] with some improvement on gastric retention. However, much work was not done on the formulation of this drug using the combined gastro retention systems. The objective of the present study was therefore to formulate and optimize sustained release floating, bioadhesive and swellable matrix tablets of ranitidine HCl using appropriate hydrophilic polymer combinations and effervescent agent.

## Materials and methods

### Materials

Ranitidine HCl (China Associate Co. Ltd, China) was kindly provided by Cadila Pharmaceutical Factory PLC, Ethiopia. Hydroxypropylmethyl cellulose (HPMC) K100M (viscosity 100,000cp, Shin-Etsu Chemicals Ltd, Japan); sodium carboxymethyl cellulose (NaCMC) and microcrystalline cellulose (MCC) (China Associate Co. Ltd, China), magnesium stearate and talc (BDH Chemicals Ltd Poole, England) and sodium bicarbonate ($NaHCO_3$) (Newport Industries Ltd, UK) were used as ingredients in the formulation development.

### Methods

**Formulation and optimization of sustained release floating, bioadhesive and swellable ranitidine HCl matrix tablets.** A Central composite design (CCD) was used for optimization of ranitidine HCl matrix tablets. Before applying the design for optimization, preliminary studies were conducted in order to identify the most critical factor variables. Factors that could possibly have significant effects on some response variables were considered including viscosity grade of polymer, polymer concentration, polymers ratio, concentration of effervescent agent and hardness of tablets. Among the evaluated factors, the concentration of polymer and concentration of effervescent agent were the most critical factors that affect the gastro-retentive and drug release characteristics. A binary mixture of HPMC/NaCMC (3:1) showed better gastro-retentive and release characteristics among the tested polymer combinations and the concentration of this polymer blend and the concentration of sodium bicarbonate were chosen as the critical formulation factors for the optimization process.

A central composite design (CCD) for 2 factors, with 5 coded values, was selected to optimize the response variables. The central point (0, 0) was studied in quintuplicate, and experimental trials were performed at all 13 possible combinations. The two formulation factors, viz., concentration of HPMC/NaCMC (3:1) ($X_1$) and concentration of $NaHCO_3$ ($X_2$) were varied as required via experimental design and the factor level were suitably coded (Table 1). Floating lag time ($Y_1$), bioadhesive strength ($Y_2$), maximum swelling index ($Y_3$), drug release at 1 h ($Y_4$), time required for 50% drug release ($t_{50\%}$) ($Y_5$) and drug release at 12 h ($Y_6$) were taken as the response variables. The resulting data were fitted into design expert software (Version 10.0.7.0, Stat- Ease Inc, Minneapolis, MN) and analyzed using analysis of variance (ANOVA). The data were also subjected to 3D response surface methodology to examine the influence of the selected formulation factors on the response variables.

The floating, bioadhesive and swellable matrix tablets of ranitidine HCl were prepared by direct compression method according to the formulations shown in Table 2. The required amount of ranitidine HCl, HPMC/NaCMC, $NaHCO_3$ and MCC were mixed thoroughly in Turbula mixer (Willy A. Bachofen AG, Turbula 2TF, Basel, Switzerland) for 10 min. Magnesium stearate and talc were then added into the powder blend and mixed for additional 10 min. Then the blend was compressed into tablets on eccentric tablet machine (EK0 Korsch, 8410–68, Berlin, Germany) fitted with 10 mm diameter flat-faced punches.

To validate the selected experimental design, the experimental values of formulation responses were quantitatively compared with those of predicted values generated by the software and the percentage relative errors were calculated.

**Evaluation of the floating, bioadhesive and swellable matrix tablets.** *Thickness*. Ten tablets were taken and the thickness of each tablet was measured using sliding caliper scale (Nippon Sokutei), Japan. Results were expressed as a mean and standard deviation.

**Table 1. Factor combination by CCD for formulation of ranitidine HCl (150mg) tablets.**

| Formulation code | Point type | Coded Factor Level | |
|---|---|---|---|
| | | $X_1$ | $X_2$ |
| F1 | Factorial | -1 | -1 |
| F2 | Factorial | +1 | -1 |
| F3 | Factorial | -1 | +1 |
| F4 | Factorial | +1 | +1 |
| F5 | Axial | $-\alpha$ | 0 |
| F6 | Axial | $+\alpha$ | 0 |
| F7 | Axial | 0 | $-\alpha$ |
| F8 | Axial | 0 | $+\alpha$ |
| F9 | Central point | 0 | 0 |
| F10 | Central point | 0 | 0 |
| F11 | Central point | 0 | 0 |
| F12 | Central point | 0 | 0 |
| F13 | Central point | 0 | 0 |

| Translation of coded levels in actual units | | | | | |
|---|---|---|---|---|---|
| Coded level | $-\alpha$ | -1 | 0 | +1 | $+\alpha$ |
| $X_1$, amount of HPMC/NaCMC (3:1) (mg) | 113.775 | 132 | 176 | 220 | 238.225 |
| $X_2$, amount of $NaHCO_3$(mg) | 39.44 | 44 | 55 | 66 | 70.556 |

$\alpha$ = 1.41421

*Hardness.* Ten tablets were taken from each batch and the crushing strengths of the tablets were determined using hardness tester (Schleuniger, 2E/205, Switzerland). Each tablet was placed between two anvils to which force was applied, and the crushing strength that just caused the tablet to break was recorded. Results were expressed as a mean and standard deviation.

*Friability.* The friability of the tablets was determined by placing 20 pre-weighed tablets in a friability tester (ERWEKA, TAR 20, Germany) and rotating them for 4 min at 25 rpm. The loss of tablet weight was calculated as a percentage of the initial weight after de-dusting the tablet.

*Floating lag time and floating duration.* The floating behavior of the tablets was determined in triplicate, according to the method described by Jimenezcastellanos et al. [16]. Briefly, a

**Table 2. Composition of floating, bioadhesive and swellable matrix tablets of ranitidine HCl (150 mg) prepared by CCD.**

| S/N | Ingredient | Formulation code and composition (mg/tab) | | | | | | | | | | | | |
|---|---|---|---|---|---|---|---|---|---|---|---|---|---|---|
| | | F1 | F2 | F3 | F4 | F5 | F6 | F7 | F8 | F9 | F10 | F11 | F12 | F13 |
| 1 | Ranitidine HCl[a] | 168 | 168 | 168 | 168 | 168 | 168 | 168 | 168 | 168 | 168 | 168 | 168 | 168 |
| 2 | HPMC/NaCMC (3:1) | 132 | 220 | 132 | 220 | 113 | 238 | 176 | 176 | 176 | 176 | 176 | 176 | 176 |
| 3 | $NaHCO_3$ | 44 | 44 | 66 | 66 | 55 | 55 | 44 | 66 | 55 | 55 | 55 | 55 | 55 |
| 4 | MCC | 118 | 30 | 96 | 8 | 126 | 1 | 74 | 52 | 63 | 63 | 63 | 63 | 63 |
| 5 | Magnesium Stearate | 4 | 4 | 4 | 4 | 4 | 4 | 4 | 4 | 4 | 4 | 4 | 4 | 4 |
| 6 | Purified Talc | 4 | 4 | 4 | 4 | 4 | 4 | 4 | 4 | 4 | 4 | 4 | 4 | 4 |
| | Total | 470 | 470 | 470 | 470 | 470 | 470 | 470 | 470 | 470 | 470 | 470 | 470 | 470 |

[a]168 mg ranitidine HCl is equivalent to 150 mg ranitidine, HPMC: hydroxypropyl methylcellulose, NaCMC: sodium carboxymethyl cellulose, NaHCO3: sodium bicarbonate, MCC: microcrystalline cellulose

tablet was placed in a glass beaker, containing 100 ml of 0.1 N HCl, maintained at 37 ± 0.5 °C. The floating lag time (the time between tablet introduction and its buoyancy) and total floating duration (the time during which tablet remains buoyant) were recorded. Results are expressed as a mean and standard deviation.

*Matrix integrity*. Matrix integrity was observed throughout *in vitro* dissolution studies and whether or not the swollen mass of the tablets remain intact was checked.

*Bioadhesive strength*. Bioadhesive strength of tablets was measured using a modified two-arm balance [17]. One holder was used to suspend the water-collecting beaker to the balance and another to suspend a glass vial to the other side of the balance. A piece of sheep stomach mucosa, 3×3 cm, obtained from a local slaughter house was used as the mucosal membrane. The mucosal membrane was separated by removing the underlying fat and loose tissues. The experiments were performed within 3 h of procurement of the mucosa. The sheep gastric mucosa was tied to an inverted 100-ml beaker and placed in a larger one (250 ml). Then, 0.1 N HCl was added into the large beaker up to the upper surface of the gastric mucosa to simulate the gastric environment. Each tablet was attached to the glass vial with adhesive, and then the beaker was raised slowly until contact between sheep mucosa and the tablet preload time were kept constant for all the formulations. In order to establish adhesion bonding between tablet and sheep stomach mucosa, a preload of 50 g was placed on the vial for a preload time of 5 min. After completion of the preload time, preload was removed from the vial and water was then added into the beaker from the burette in the other side. The addition of water was stopped when the tablet was detached from the sheep mucosa. The weight of water required to detach the tablet from the mucosa was noted as mucoadhesive strength. The experiment was done in triplicate and mean and standard deviation were calculated.

*Tablet adhesion retention period (ex vivo mucoadhesion time)*. The *ex vivo* adhesion retention time of the tablets was determined using a locally modified USP tablet disintegration test apparatus (2T 504, Erweka, Germany), based on the method reported by Nakamura et al. [18]. The medium was composed of 800 ml 0.1N HCl (pH 1.2) in 1 L glass beaker maintained at 37 ± 1 °C. A segment of sheep gastric mucosa, 2.5 × 2.5 cm, obtained from a local slaughter house, was cut and glued (with cyanoacrylate adhesive) to the surface of a glass slide. One side of the tablet was wetted with 50 μL of 0.1 N HCl and attached to the center of the sheep stomach mucosa by applying a light force with a fingertip for 20 s. The glass was vertically fixed to the apparatus, 5 min later, and allowed to move up and down (25 times per min) so that the tablet was completely immersed in the solution at the lowest point and was out at the highest point. The time required for the tablet to detach from the gastric mucosa was recorded as the mucoadhesion time. The experiment was repeated thrice and the average was taken.

*Swelling index and determination of diameter of swollen tablets*. The swelling behavior of the tablets was determined, in triplicate, according to the method described by Dorozynski et al. [19]. In this, the tablets were weighed individually (designated as $W_0$) and placed separately in glass beaker containing 200 ml of 0.1 N HCl maintained at 37 °C ± 0.5 °C. The tablets were removed from the beaker, and the excess surface liquid was removed carefully using tissue paper at regular time intervals, up to 12 h. The swollen tablets were then re-weighed ($W_t$), and % swelling index (SI) was calculated (Eq 1). The experiment was done in triplicate and mean and standard deviation were calculated.

$$SI\ (\%) = \frac{Wt - Wo}{Wo} x100 \tag{Eq1}$$

Where, SI is swelling index, $W_t$ is weight of swollen tablet at time t, $W_0$ is initial weight of the tablet. The diameter of swollen tablets was measured at 1 h and 12 h, in triplicate, using ruler. The mean and standard deviation were calculated.

*In vitro drug release*. The *in vitro* drug release studies were performed using USP type II dissolution apparatus (Pharma test type PTDTT, Germany) at 50 rpm in 900 ml of 0.1 N HCl dissolution medium at 37 ± 0.5˚C. Aliquot samples of 5 ml were withdrawn from each dissolution vessel after pre scheduled time intervals (0.25, 0.50, 1, 2, 3, 4, 6, 8, 10, and 12 h) and replaced with an equal volume of fresh dissolution medium which was kept at 37 ± 0.5˚C to maintain sink condition. The samples were filtered through a filter paper of 0.45 μm size and were then analyzed spectrophotometrically (T92+ Spectrophotometer, PG Instruments Ltd., UK) at maximum wavelength of 314 nm (USP 36/NF31, 2013).

*Drug content analysis*. Twenty tablets were weighed and finely powdered. An accurately weighed powder of ranitidine HCl equivalent to 100 mg of ranitidine was dissolved in about 60 ml of 0.1N HCl. The solution was diluted to 100 ml with 0.1N HCl and filtered. 5 ml of the filtrate was further diluted to 50 ml with the same solvent. The absorbance of the resultant solution was measured spectrophotometrically (T92+ Spectrophotometer, PG Instruments Ltd., UK) at 314 nm using 0.1 N HCl as blank and the content was calculated from the absorbance of sample and standard. The experiment was done in triplicate for each batch and mean and standard deviation were calculated.

**Analysis of kinetics and mechanism of drug release.** The dissolution data were fitted into the different drug release kinetic models (zero order, first order, Higuchi, Hixson-Crowell, and Korsmeyer–Peppas model) to evaluate the rate and mechanism of drug release from the matrix tablets [20]. The order and mechanism of drug release from the matrix system were determined based on regression ($R^2$) values [21].

**Statistical analysis.** One-way analysis of variance (ANOVA) and Origin Pro 8.5.1 were applied for comparison of results. To demonstrate graphically the influence of each factor on responses and to indicate the optimum level of factors, the contour and response surface plots were generated using Design-Expert 10.0.7.0 software (Stat-ease, Corp. Australia). At 95% confidence interval, p-values of $< 0.05$ were considered statistically significant. All the data measured and reported were averages of a minimum of triplicate measurements and the values are expressed as mean ± standard deviation.

## Results & discussion

### Characteristics of tablets

The physico-chemical and gastro-retentive characteristics of the different formulations are given in Table 3. The tablets mean thickness values ranged from 3.89 mm to 4.02 mm. The hardness of the tablets ranged between 60.4–67.8 N. The friability was in a range of 0.18–0.50% and this was within the pharmacopeia limit ($< 1\%$). The drug content of tablets was 98.25–102.50% and this was also within the limit (90–110) %.

The tablets exhibited floating duration of more than 12 h and the *ex vivo* mucoadhesion time of more than 12 h except for formulations $F_1$, $F_3$ and $F_5$. The *in vitro* swelling study showed that all batches had good swelling properties. The matrix integrity of all the tablets was maintained for up to 12 h; and the average diameter after 1 h and 12 h were above 14 mm (greater than the average diameter of the pylorus sphincter, 12.8 mm) for all the formulations except for formulation $F_5$. The swelling index profiles of the 13 formulations are shown in Fig 1. The raw data for Table 3 and Fig 1 are stated in the supplementary data (S1 and S2 Tables, respectively).

The mean average experimental results for the selected response variables are given separately in Table 4 and individual results are mentioned in S3 Table. The floating lag time ranged from 4.10–13.54 sec. An increase in the concentration of polymer showed decrease in floating lag time. For example, the floating lag time of formulations $F_5$ (24.21% polymer), $F_{12}$ (37.40%

Table 3. Characteristic properties of floating, bioadhesive and swelling sustained release matrix tablets of ranitidine HCl (150).

| Formulation Code | Hardness (N) | Friability (%) | Thickness (mm) | Assay (%) | Floating duration (h) | Matrix integrity | *Ex vivo* mucoadhesion time (h) | Average diameter | |
|---|---|---|---|---|---|---|---|---|---|
| | | | | | | | | 1h (mm) | 12 h (mm) |
| F1 | 62.2 ± 0.01 | 0.31 | 3.89 ± 0.00 | 98.25 ± 0.04 | > 24 | + | 10.50 | 14.08 ± 0.01 | 14.56 ± 0.01 |
| F2 | 65.4 ± 0.01 | 0.41 | 3.95 ± 0.01 | 101.01 ± 0.01 | > 24 | + | >12 | 14.14 ± 0.03 | 14.78 ± 0.01 |
| F3 | 62.1 ± 0.08 | 0.32 | 3.95 ± 0.04 | 102.50 ± 0.01 | > 24 | + | 10.05 | 14.06 ± 0.01 | 14.63 ± 0.01 |
| F4 | 63.1 ± 0.01 | 0.42 | 3.91 ± 0.04 | 100.01 ± 0.00 | > 24 | + | >12 | 14.52 ± 0.01 | 14.86 ± 0.01 |
| F5 | 64.4 ± 0.08 | 0.50 | 3.95 ± 0.00 | 99.56 ± 0.04 | > 24 | NI[a] | 6.25 | 13.32 ± 0.04 | 13.95 ± 0.01 |
| F6 | 61.5 ± 0.04 | 0.41 | 4.01 ± 0.05 | 98.87 ± 0.09 | > 24 | + | >12 | 14.73 ± 0.01 | 15.02 ± 0.01 |
| F7 | 67.8 ± 0.04 | 0.41 | 3.98 ± 0.01 | 100.01 ± 0.04 | > 24 | + | >12 | 14.19 ± 0.01 | 14.61 ± 0.00 |
| F8 | 66.4 ± 0.05 | 0.42 | 4.01 ± 0.01 | 101.04 ± 0.00 | > 24 | + | >12 | 14.36 ± 0.01 | 14.73 ± 0.01 |
| F9 | 60.4 ± 0.01 | 0.31 | 4.00 ± 0.04 | 102.00 ± 0.00 | > 24 | + | >12 | 14.24 ± 0.02 | 14.91 ± 0.01 |
| F10 | 64.1 ± 0.00 | 0.51 | 3.98 ± 0.00 | 98.68 ± 0.04 | > 24 | + | >12 | 14.20 ± 0.01 | 14.86 ± 0.04 |
| F11 | 65.4 ± 0.03 | 0.21 | 3.97 ± 0.01 | 99.76 ± 0.05 | > 24 | + | >12 | 14.27 ± 0.01 | 14.85 ± 0.05 |
| F12 | 61.5 ± 0.02 | 0.18 | 4.02 ± 0.03 | 100.21 ± 0.02 | > 24 | + | >12 | 14.31 ± 0.01 | 14.87 ± 0.04 |
| F13 | 63.7 ± 0.00 | 0.24 | 4.00 ± 0.01 | 99.27 ± 0.01 | > 24 | + | >12 | 14.29 ± 0.10 | ± 0.03 |

[a]Not intact

polymer) and $F_6$ (50.69% polymer) were 8.60, 7.50 and 6.44 sec, respectively ($p < 0.05$). This might be due to the faster swelling rate and rapid formation of thick gel around the tablets at higher polymer concentration causing rapid entrapment of generated $CO_2$ [3, 22]. The gas generated is trapped and protected within the gel formed by hydration of the polymer, thus lowering the density of the tablet below that of gastric contents, causing buoyancy or floatation. On the other hand, the floating lag time decreased with increasing amount of sodium bicarbonate. For instance, the floating lag time of formulations $F_7$ (8.40% $NaHCO_3$), $F_9$ (11.70% $NaHCO_3$), and $F_8$ (15.01% $NaHCO_3$) were 13.54, 6.20, and 4.10 sec ($p < 0.05$),

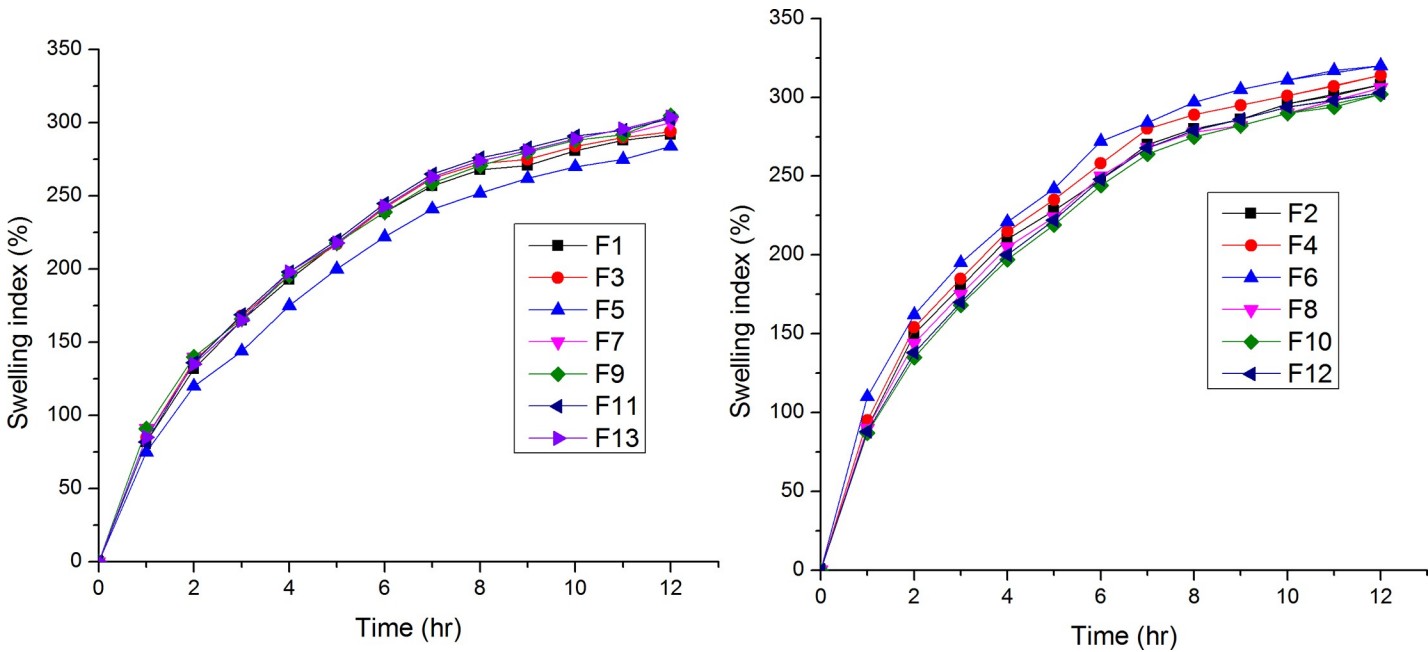

Fig 1. Swelling index profiles of ranitidine HCl matrix tablets.

**Table 4. Formulations of ranitidine HCl (150 mg) matrix tablets with the levels of independent variables and observed values for the response variables.**

| Formulation | Independent variables | | Observed response | | | | | |
|---|---|---|---|---|---|---|---|---|
| | $X_1$ | $X_2$ | $Y_1$ (sec) | $Y_2$ | $Y_3$ | $Y_4$ | $Y_5$ | $Y_6$ |
| | (mg) | (mg) | | (g) | (%) | (%) | (%.h$^{-1/2}$) | (%) |
| F1 | 132 | 44 | 12.25±0.04 | 23.80±0.29 | 292±1.08 | 30.08±0.19 | 3.40±0.65 | 98.50±1.08 |
| F2 | 220 | 44 | 11.28±0.12 | 30.35±0.64 | 308±0.25 | 22.40±0.24 | 3.91±0.24 | 87.60±0.58 |
| F3 | 132 | 66 | 6.20±0.32 | 23.30±0.45 | 294±0.65 | 30.50±0.25 | 3.23±0.57 | 98.00±0.78 |
| F4 | 220 | 66 | 5.10±0.45 | 29.47±0.25 | 314±0.14 | 23.10±078 | 3.84±0.62 | 88.90±0.35 |
| F5 | 113 | 55 | 8.50±0.21 | 20.23±0.65 | 284±0.54 | 31.50±0.35 | 3.15±0.35 | 100.00±0.65 |
| F6 | 238 | 55 | 6.44±0.24 | 32.50±0.47 | 320±0.35 | 21.10±0.64 | 3.98±0.29 | 87.85±0.28 |
| F7 | 176 | 44 | 13.54±0.57 | 27.24±0.32 | 300±0.64 | 24.10±0.52 | 3.83±0.27 | 92.30±0.47 |
| F8 | 176 | 66 | 4.10±0.68 | 24.40±0.45 | 306±0.71 | 28.31±0.28 | 3.50±0.39 | 95.30±0.29 |
| F9 | 176 | 55 | 6.20±0.52 | 26.43±0.65 | 305±0.82 | 26.35±0.49 | 3.85±0.60 | 94.80±0.37 |
| F10 | 176 | 55 | 6.50±0.35 | 26.57±1.08 | 302±0.25 | 25.02±0.67 | 3.75±0.28 | 94.30±0.46 |
| F11 | 176 | 55 | 7.00±0.41 | 27.10±0.87 | 303±0.23 | 25.50±0.39 | 3.84±0.34 | 93.65±0.49 |
| F12 | 176 | 55 | 7.50±0.75 | 25.50±0.49 | 303±0.54 | 26.21±0.28 | 3.75±0.19 | 93.50±0.85 |
| F13 | 176 | 55 | 6.60±0.35 | 26.57±1.07 | 304±0.68 | 27.04±0.52 | 3.79±0.12 | 93.50±0.54 |

$X_1$: HPMC/NaCMC (3:1) (mg), $X_2$: NaHCO$_3$ (mg), $Y_1$: Floating lag time $Y_2$: Bioadhesive strength, $Y_3$: Maximum swelling index, $Y_4$: Cumulative drug release at 1 h, $Y_5$: $t_{50\%}$, $Y_6$: Cumulative drug release at 12 h.

respectively. The amount of $CO_2$ produced is exclusively proportional to the quantity of NaHCO$_3$ in the tablet [23]. Hence, decrease in floating lag time can be attributed to the availability of an increased amount of $CO_2$ as the concentration of NaHCO$_3$ was increased, and the gas is entrapped in the formed gel to give rapid buoyancy.

All formulations showed sufficiently high bio-adhesion strength ($> 20$ g) (Table 4). The bioadhesive strength ranged from 20.23–32.50 g. A distinct increase in the bioadhesive strength was observed with an increase in the amount of polymer. As shown in Table 4, formulations $F_5$ (24.21% polymer), $F_{10}$ (37.40% polymer) and $F_6$ (50.69% polymer) have bioadhesive strength of 20.23, 26.57 and 32.50 g, respectively ($p < 0.05$). This increase could be due to the nature of the polymers. NaCMC is an anionic polymer and contains proton donating carboxyl groups responsible for its higher mucoadhesive strength [24]. On the other hand, HPMC is a long-chained, non-ionic polymer and has a bioadhesive property due to formation of physical or hydrogen bonding with the mucus components [9]. Hence, the increase in bioadhesive strength with increase in polymer concentration is attributed to the rapid swelling of hydrogels when in contact with hydrated mucous membrane, resulting in reduced glass transition temperature and increased uncoiling along with an increased mobility of polymer chains [17]. This tends to increase the adhesive surface for maximum contact with mucin and flexibility for interpenetration with mucin.

There was a decrease in bioadhesive strength with an increase in concentration of sodium bicarbonate. Formulations $F_7$ (8.40% NaHCO$_3$), $F_9$ (11.70% NaHCO$_3$), and $F_8$ (15.01% NaHCO$_3$) exhibited bioadhesive strength of 27.24, 26.43 and 24.40g, respectively ($p < 0.05$). Previous study by Kotagale et al [25] also reported the same effect. The decrease in bioadhesive property might be attributed to the formation of higher $CO_2$ bubbles on the tablet surface in formulations having higher amount of NaHCO$_3$. Moreover, increasing sodium bicarbonate concentration increases the micro-environment pH and water uptake and decreases the bioadhesive strength [25].

The results of the swelling index (Fig 1 and Table 4) indicate that swelling index of the tablets increased with increase in polymer concentration and effervescent agent. The swelling index was highest for formulation $F_6$ (50.69% polymer) and least for $F_5$ (24.21% polymer).

This is attributed to the presence of highly hydrophilic hydroxypropyl groups in HPMC [26, 27]. The hydration of these functional groups results in rapid ingress of medium in polymer network, ultimately causing swelling and ordering of polymer chains [26]. In the same manner, comparing formulations $F_7$ (8.40% $NaHCO_3$), $F_{12}$ (11.70% $NaHCO_3$), and $F_8$ (15.01% $NaHCO_3$), $F_8$ had the highest swelling index. This could be explained by the disintegration effect of the effervescent agent, which increases the volume of the tablets and porous channels on the surface and the inner part of the tablets. The porous channels increase the contact area between polymer particles and water so that the polymers could be hydrated more easily.

As can be seen from Table 4, the drug release in the first 1 h ranged from 21.10–31.50%. At the end of 12 h, the value ranged between 87.60 and 100%. The $t_{50\%}$ varied between 3.15 and 3.98%. $h^{-1/2}$. Sustained release up to 12 h was achieved in all formulations except $F_5$ in which about 100% of the drug release was observed within 8 h (Fig 2 or S4 Table). For the majority of the formulations, the rate of drug release was found to be faster during the first 2 h followed by a gradual release phase for more than 10 h. The initial faster release may be due to surface erosion and shorter diffusion path. The release rate then decreases as the external layers of the tablet deplete and therefore, drug must travel longer path through the gel layer to reach the surface. The *in vitro* drug release profile revealed that formulations with lower polymer concentration exhibited initial burst drug release.

Drug release rate was slower at higher concentration of polymer. This could be attributed to the effect of polymer on the tortuosity of the gel and the formation of a strong gel layer that occurs with high polymer contents. Furthermore, the diffusion layer becomes stronger and more resistant to diffusion and erosion as the polymer concentration is increased [28]. Moreover, drug release was faster with an increase in concentration of $NaHCO_3$. As can be seen in Table 4, the drug release after 12 h for formulations F7 (8.40% $NaHCO_3$), F12 (11.70% $NaHCO_3$), and F8 (15.01% $NaHCO_3$) were 92.30%, 93.50% and 95.30%, respectively (p < 0.005). The raw data for all the above experiments are attached in supplementary data (S1–S5 Tables).

### Drug release kinetics

The determination coefficient ($R^2$) was used as an indicator of the best fitting for each of the models considered. Among the different models studied, the dissolution data best fitted the Higuchi model; drug release was proportional to the square root of time.

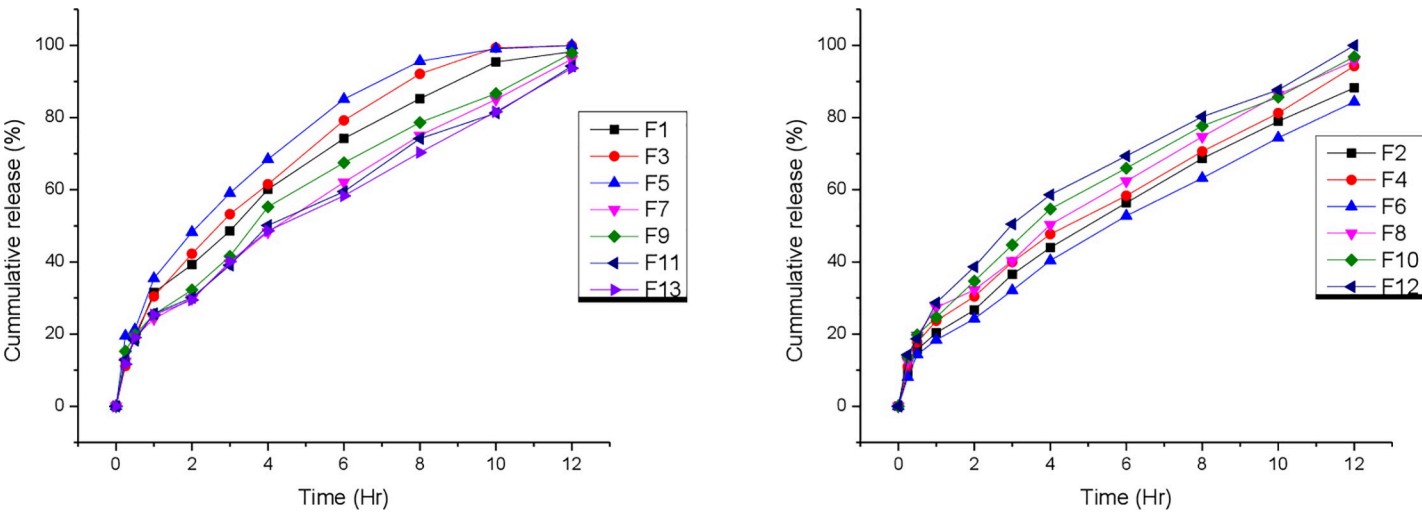

**Fig 2. In vitro release profiles of the 13 formulations of ranitidine HCl (150 mg) matrix.**

The drug release mechanism was determined by fitting the data from the first 60% drug release to the Korsmeyer-Peppas model. In this model, the value of the diffusion constant (n) illustrates the type of release mechanism. For cylindrical tablets; $n \leq 0.45$ corresponds to a Fickian diffusion mechanism, $0.45 < n < 0.89$ to non-Fickian transport, $n = 0.89$ to Case II (relaxational) transport, and $n > 0.89$ to super case II transport. As shown in S5 Table, the value of diffusion exponent (n) ranged from 0.4722 to 0.6045 indicating an anomalous diffusion ($0.45 < n < 0.89$) mechanism. Anomalous diffusion of drug release mechanism signifies a coupling of both diffusion and erosion mechanisms which indicates that the drug release is controlled by more than one process during the entire period of drug release [20].

## Optimization

Central composite design was used to investigate the effect of the two independent variables and their potential interaction. The average values (shown in Table 4) were submitted to multiple regression analysis using Design Expert® software (Version 10.0.7.0, Stat- Ease Inc, Minneapolis, MN). Polynomial models were generated for all the response variables. The best fitting mathematical model was selected based on the comparisons of several statistical parameters including the coefficient of variation (CV), the multiple correlation coefficient ($R^2$), adjusted multiple correlation coefficient (adjusted $R^2$) and the predicted residual sum of square (PRESS) (S6 Table).

The fit of the model was evaluated using $R^2$-values. As observed from S2 Table, $R^2$ was $> 0.95$ for all responses, which indicates a high degree of correlation between the experimental and predicted responses. In addition, the 'Predicted $R^2$' value was in good agreement with the 'Adjusted $R^2$' value (the difference is less than 0.2), indicating the reliability of the models. It was also shown that the selected model for each response variable have lower PRESS values compared to other models, again, indicating the fitness of the models.

Based on the fit summary shown in S6 Table, linear model was selected as best fit for bioadhesive strength, swelling index, drug release at 1 h and 12 h; and the quadratic model was selected as best fit for floating lag time and t50%, as suggested by the software. The reliability of the selected models was verified by using the analysis of variance (ANOVA) and other statistical parameters. The results of ANOVA (S7 Table) indicated that the selected models were significant ($p < 0.05$) and the lack of fit (LOF) were not significant (p-values $> 0.05$) for all the response variables indicating the reliability of the models.

The results of statistical analysis for the selected mathematical models are shown in Table 5. The 'Predicted R-Squared' value is in good agreement with the 'Adjusted R-Squared' value (difference $< 0.2$), indicating reliable models. Adequate precision (which measures signal to noise ratio) was greater than 4 for all the responses showing that the proposed models can be

**Table 5.  Numerical test results of model adequacy checking for influence of independent variables on response variables.**

| Response | Source | R-squared | Adjusted R-squared | Predicted R-squared | Adequate precision | % CV |
|----------|--------|-----------|--------------------|--------------------|--------------------|------|
| $Y_1$ | Quadratic | 0.9793 | 0.9646 | 0.9103 | 24.880 | 6.92 |
| $Y_2$ | Linear | 0.9530 | 0.9436 | 0.9081 | 29.17 | 2.83 |
| $Y_3$ | Linear | 0.9583 | 0.9499 | 0.9157 | 30.99 | 0.68 |
| $Y_4$ | Linear | 0.9515 | 0.9418 | 0.9157 | 28.37 | 2.94 |
| $Y_5$ | Quadratic | 0.9806 | 0.9668 | 0.9209 | 24.45 | 1.33 |
| $Y_6$ | Linear | 0.9672 | 0.9606 | 0.9370 | 35.40 | 0.82 |

used to navigate the design space. Furthermore, for all the models, % CV were $< 10\%$ indicating the reproducibility of the selected model.

In order to determine the levels of factors which yield optimum responses, mathematical relationships were generated between the dependent and independent variables. Polynomial equations for responses which comprised the coefficients for intercept, first-order main effects, interaction terms, and higher order effects were derived. The final equations in terms of coded factors are shown in Eq 2 through Eq 7.

$$Y_1 = +6.76 - 0.63 * X_1 - 3.21 * X_2 - 0.03 * X_1 X_2 + 0.49 * X_1{}^2 + 1.15 * X_2{}^2 \qquad \text{Eq2}$$

$$Y_2 = +26.40 + 3.71 * X_1 - 0.67 * X_2 \qquad \text{Eq3}$$

$$Y_3 = +302.69 + 10.86 * X_1 + 2.06 * X_2 \qquad \text{Eq4}$$

$$Y_4 = +26.25 - 3.72 * X_1 + 0.88 * X_2 \qquad \text{Eq5}$$

$$Y_5 = +3.80 + 0.29 * X_1 - 0.088 * X_2 + 0.025 * X_1 X_2 - 0.12 * X_1{}^2 - 0.071 * X_2{}^2 \qquad \text{Eq6}$$

$$Y_6 = +93.70 - 4.63 * X_1 + 0.63 * X_2 \qquad \text{Eq7}$$

The values obtained for the main effects of each factor in Eqs 2 – 7 reveal that $X_1$ (HPMC/NaCMC (3:1)) individually, has rather more pronounced effect on the $Y_2$ (bioadhesive strength), $Y_3$ (maximum swelling index), $Y_4$ (cumulative drug release at 1 h), $Y_5$ (t50%) and $Y_6$ (cumulative drug release at 12 h); while X2 (NaHCO$_3$) has more pronounced effect on $Y_1$ (floating lag time). The influence of the two independent variables on each of the six response variables was also elucidated by response surface plots (S1 Fig).

Certain gastro-retentive and release criteria were set in order to determine the optimal concentrations of HPMC/NaCMC (3:1) and NaHCO$_3$. Minimizing floating lag time, maximizing bioadhesive strength and attaining sufficient swelling and drug release properties were set as goal in the optimization process.

The desirability function approach is one of the most widely used methods for optimization of multiple responses. Overall desirability function is a measure of how well the combined goals for all responses are satisfied. Desirability function ranges from 0 to 1, with value closer to one indicating a higher satisfaction of response goal(s). The numerical optimization tool provided 9 sets of optimal solutions among which 214.55 mg of HPMC:NaCMC (3:1) and 61.22 mg of NaHCO$_3$ was selected (by the software) as optimized concentration with desirability of 0.856. The area of optimized formulation was also ratified using overlay plot as shown in Fig 3 in which the yellow region represents the area satisfying the imposed criteria.

Experiments were carried out in triplicate at the selected optimum concentrations (214.55 mg of HPMC:NaCMC (3:1) and 61.22 mg of NaHCO3) and the resulting matrix tablets were evaluated for gastroretentive, release and other physicochemical properties. The tablets showed low friability (0.25%), acceptable drug content value (99.06 ± 0.00) and were intact. and floated for more than 12 h with *ex vivo* mucoadhesion time of $> 12$ h. The floating lag time was 5.09 sec and the bio adhesive strength was 29.69 g. The diameter of the swollen tablets after 1 h and 12 h were 14.26 mm and 14.85 mm, respectively. *In vitro* dissolution study was carried out on the optimized formulation for three different batches (Fig 4 and S8 Table).

The results of *in vitro* dissolution study indicate that there was no significant difference in the release properties among the three batches; suggesting that the optimal formulation had reproducible drug release properties.

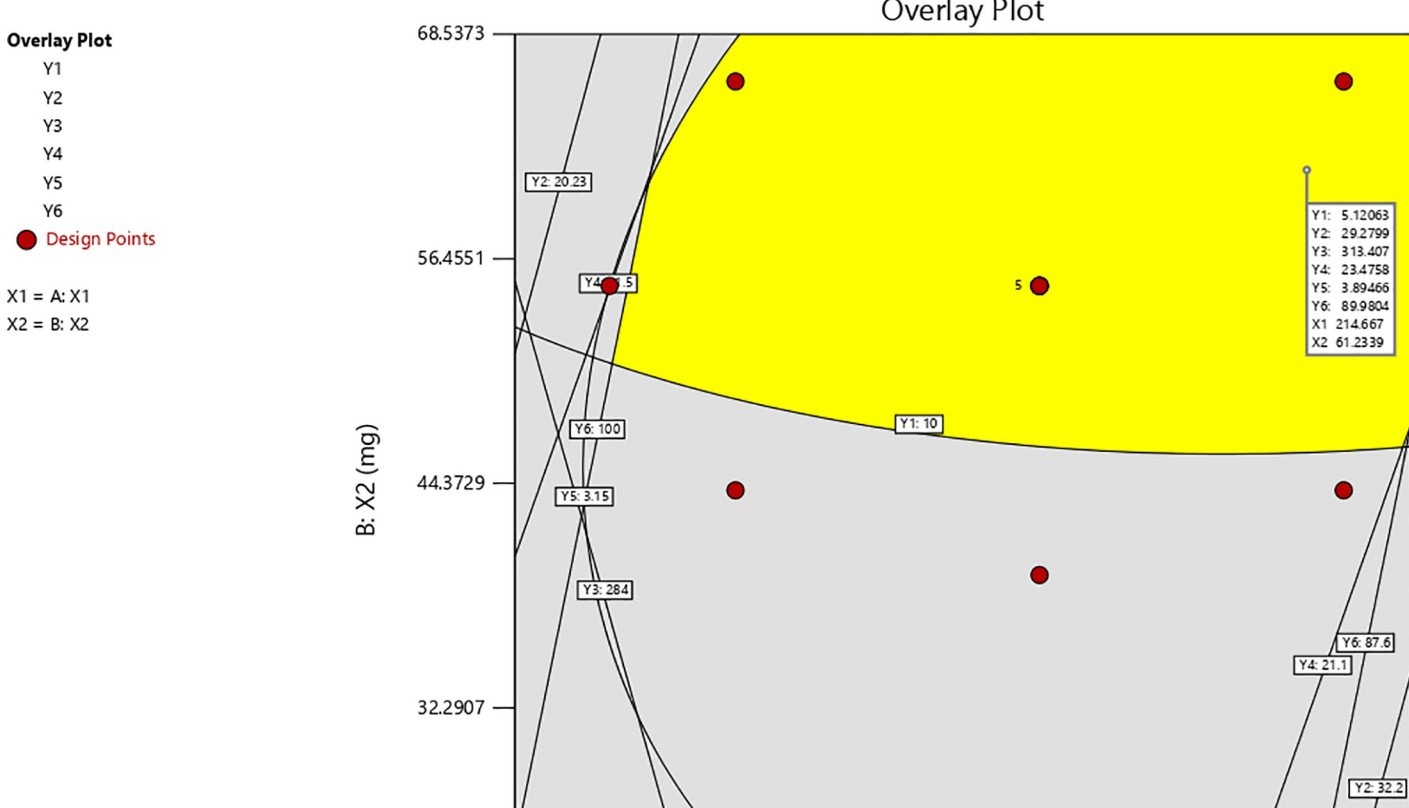

**Fig 3. Overlay plot of optimized formulation of ranitidine HCl matrix tablets.**

To confirm the validity of obtained optimal formulation, experiments were carried out in triplicate at the optimal combinations of the factors (X1 = 214.55 mg, X2 = 61.22 mg). Table 6 provides the predicted values and experimental results for the response variables and the percentage error obtained at optimal levels. The observed values of response variables were close to the predicted values (error < 5%), indicating the reliability of developed mathematical models.

Release kinetics study on the optimized formulation revealed that Higuchi square root model was the best fit model with $R^2 > 0.996$. The drug release mechanism from the optimized formulation was also evaluated using the Korsmeyer-Peppas model at 60% release and the results showed that n value ranges from 0.55 to 0.58 indicating drug release from the optimized formulation follows non-Fickian diffusion release mechanism.

The floating, bioadhesive, swelling and *in-vitro* drug release characteristics of the optimized formulation is summarized in S2 Fig. As can be seen from the graphical presentation, the matrix tablet has the desired gastro-retentive and release characteristics.

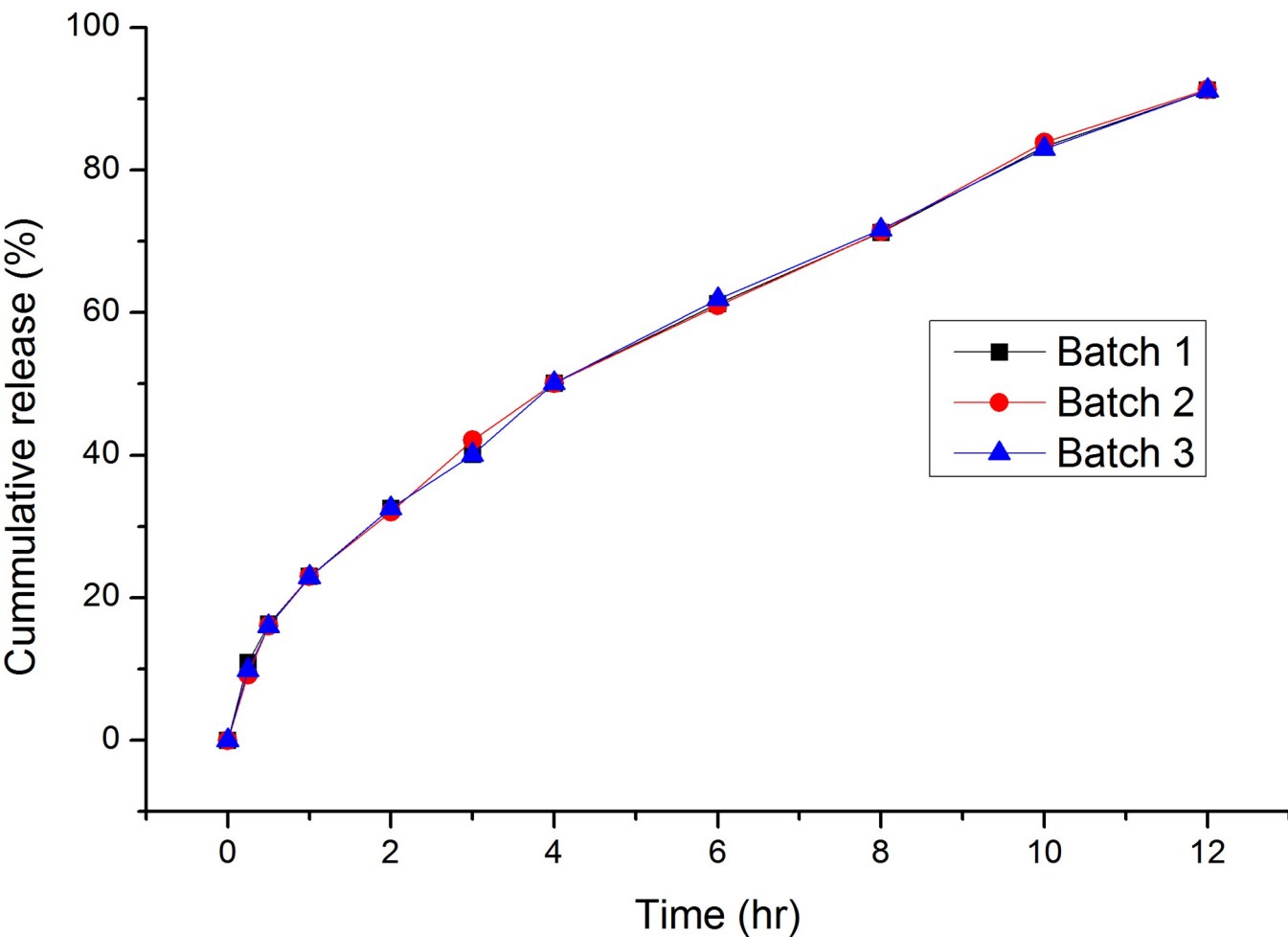

**Fig 4. Release profile of optimized formulation of ranitidine HCl matrix tablets.**

## Conclusion

In this study, ranitidine HCl matrix tablets having a combination of gastroretentive characteristics (floating, bioadhesive and swelling) were prepared using central composite design. The optimized formulation sustained the drug release for more than 12 h. Thus, this approach can be considered as effective gastroretentive strategy to deliver the drug at the absorption site and thereby improving its bioavailability.

**Table 6. Response values of predicted, experimental and percentage error obtained at optimal levels of the factors.**

| Response | Predicted value | Experimental value | % Error |
|---|---|---|---|
| FLT (sec) ($Y_1$) | 5.12 | 5.09 | -0.59 |
| Bioadhesive strength (g) ($Y_2$) | 29.27 | 29.69 | 1.30 |
| Max. swelling index (%) ($Y_3$) | 313.37 | 315.04 | 0.53 |
| Release at 1h (%) ($Y_4$) | 23.48 | 24.21 | 3.11 |
| t50% (h) ($Y_5$) | 3.89 | 3.86 | -0.77 |
| Release to 12 h (%) ($Y_6$) | 90.00 | 93.65 | 4.06 |

## Supporting information

**S1 Fig.** Response surface plots of the 13 formulations of ranitidine HCl matrix tablets showing the effect of concentration of HPMC/NaCMC (3:1) ($X_1$) and Sodium Bicarbonate ($X_2$) on a) floating lag time ($Y_1$), b) bioadhesive strength ($Y_2$), c) swelling index ($Y_3$), d) cumulative drug release at 1 h ($Y_4$), e) $t_{50\%}$ ($Y_5$) and f) drug release at 12 h ($Y_6$).
(TIFF)

**S2 Fig. Gastro-retentive and release characteristics of the optimized formulation of sustained release floating, bioadhesive and swelling matrix tablets of Ranitidine HCl.**
(TIFF)

**S1 Table. Rate constants and correlation coefficient fits of different kinetic equations for the 13 formulations of ranitidine HCl (150 mg) matrix tablets prepared as per CCD.**
(DOCX)

**S2 Table. Fit summary statistics for response variables of the 13 formulations of ranitidine HCl (150mg) matrix tablets prepared as per CCD.**
(DOCX)

**S3 Table. Summary of ANOVA showing influence of independent variables on response variables.**
(DOCX)

**S4 Table. Raw data for *in vitro* release profile of 13 formulations of ranitidine HCl (150 mg) matrix tablets (raw data used to plot Fig 2).**
(DOCX)

**S5 Table. Rate constants and correlation coefficient fits of different kinetic equations for the 13 formulations of ranitidine HCl (150 mg) matrix tablets prepared as per CCD.**
(DOCX)

**S6 Table. Fit summary statistics for response variables of the 13 formulations of ranitidine HCl (150mg) matrix tablets prepared as per CCD.**
(DOCX)

**S7 Table. Summary of ANOVA showing influence of independent variables on response variables.**
(DOCX)

**S8 Table. Raw data for *in vitro* release profile of optimized formulation of ranitidine HCl (150 mg) matrix tablets (raw data used to plot Fig 4).**
(DOCX)

## Author Contributions

**Conceptualization:** Birhanu Nigusse, Tsige Gebre-Mariam.

**Supervision:** Tsige Gebre-Mariam, Anteneh Belete.

**Writing – original draft:** Birhanu Nigusse.

**Writing – review & editing:** Birhanu Nigusse, Tsige Gebre-Mariam, Anteneh Belete.

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
