## [Decision Letter · Decision Letter 0]

10 Mar 2021

PONE-D-20-37614

Design, Development and Optimization of Sustained Release Floating, Bioadhesive and Swellable Matrix Tablet of Ranitidine Hydrochloride

PLOS ONE

Dear Dr. Gebre-Mariam,

Thank you for submitting your manuscript to PLOS ONE. After careful consideration, we feel that it has merit but does not fully meet PLOS ONE’s publication criteria as it currently stands. Therefore, we invite you to submit a revised version of the manuscript that addresses the points raised during the review process.

We look forward to receiving your revised manuscript.

Kind regards,

Vineet Kumar Rai, PhD

Academic Editor

PLOS ONE

Journal Requirements:

Reviewers' comments:

Reviewer's Responses to Questions

**Comments to the Author**

1. Is the manuscript technically sound, and do the data support the conclusions?

Reviewer #1: Yes

Reviewer #2: Yes

2. Has the statistical analysis been performed appropriately and rigorously? 

Reviewer #1: Yes

Reviewer #2: Yes

3. Have the authors made all data underlying the findings in their manuscript fully available?

Reviewer #1: Yes

Reviewer #2: Yes

4. Is the manuscript presented in an intelligible fashion and written in standard English?

Reviewer #1: Yes

Reviewer #2: Yes

5. Review Comments to the Author

Reviewer #1: 1. I read this manuscript and I think that the paper is well written. Excellent contribution in terms of content and overall descriptions. The authors well described the Formulation development and evaluation of floating tablet. The tables and figures are good although I have some concerns about the resolution of the figures.

2. I suggest the authors to go through the whole content thoroughly to avoid any phrase and alignment problems and grammatical error that may influence the convey of idea.

3. please mention the Pos hoc test performed along with ANOVA in statistical analysis section.

4. Please mention at what point NaHCO3 was added during tablet formulation in methodology section.

5. Can author explained the relevance of ex vivo mucoadhesion time as formulation is floating type.

6. What is the rationale to do swelling test till 12 hrs. Discuss in detail regading effect of polymer and NaHCO3 concentration on swelling and bioadhesion with relevant references of similar work..

7. Include standard deviation in table 4.

8. Elaborate drug release discussion with relevant references.

9. Author are suggest to cite more recent and relevant references.

Reviewer #2: Manuscript Number: PONE-D-20-37614

Article Type: Research Article

Full Title: Design, Development and Optimization of Sustained Release Floating, Bioadhesive

and Swellable Matrix Tablet of Ranitidine Hydrochloride

Dear Authors

The present study was to attempt a SR tablet of Ranitidine based on Bio-adhesion mechanism. Study seems to be OK., but not in detailed layout., which was questionable to be published in Plos One. However, results were reported in GOOD state. The final decision will be taken by Eic. I am writing my comments below:

In Introduction, the actual biopharmaceutical problems of Ranitidine were missing. What is the rationale of preparing its swellable tablet should be mentioned in the Introduction

In CCD Design, α-value and design space for the developed formulations?

The polynomial equations mentioned by the authors were not in quadratic state, as mentioned in statistical analysis

PRESS value seems to be higher in the mathematical models? A justification is needed

What is desirability value of optimized formulation? A graphical optimization graph should be provided?

Major characterization studies have been done by the Authors, however a factual discussion is missing

What is the rationale of doing the release profile of preliminary batches of CDD?

What is the floatation behavior of optimized formulation

Overall, the study is not novel, but the presentation is OK

6. PLOS authors have the option to publish the peer review history of their article (what does this mean?). If published, this will include your full peer review and any attached files.

Reviewer #1: No

Reviewer #2: **Yes: **ok

---

## [Author Response · Author response to Decision Letter 0]

25 Mar 2021

Rebuttal letter

We thank the editor and the two reviewers for their comments on our manuscript titled “Design, Development and Optimization of Sustained Release Floating, Bioadhesive and Swellable Matrix Tablet of Ranitidine Hydrochloride (PONE-D-20-37614)”. The comments have significantly improved the quality of the manuscript.

Below is our response, point-by-point, to comments forwarded by the academic editor and the reviewers.

Sincerely, 

Prof. Tsige Gebre-Mariam 

On behalf of the authors

Academic editor 

Authors’ response: PLOS ONE’s style requirements including are now fulfilled, those for file naming. 

2. PLOS requires an ORCID ID for the corresponding author in Editorial Manager on papers submitted after December 6th, 2016. Please ensure that you have an ORCID iD and that it is validated in Editorial Manager.

Authors’ response: The corresponding author has ORCID ID and it is validated in Editorial Manager. 

3. Authors’ response: We have made some changes in our financial disclosure section of the online submission and the updated statement is as follows:

BN would like to thank the Regional Bioequivalence Center for sponsoring his MSc study, and School of Pharmacy, College of Health Sciences, Addis Ababa University for the financial and material support. BN would also like to thank Cadila Pharmaceutical Factory PLC for material support. 

4. While revising your submission, please upload your figure files to the Preflight Analysis and Conversion Engine (PACE) digital diagnostic tool, https://pacev2.apexcovantage.com/

Authors’ response: The figure files have been uploaded to the PACE and they now fulfill PLOS requirements. 

Reviewer #1

1. I read this manuscript and I think that the paper is well written. Excellent contribution in terms of content and overall descriptions. The authors well described the Formulation development and evaluation of floating tablet. The tables and figures are good although I have some concerns about the resolution of the figures.

Authors’ response: The resolution of the figures is now improved and it fulfils PLOS requirements. 

2. I suggest the authors to go through the whole content thoroughly to avoid any phrase and alignment problems and grammatical error that may influence the convey of idea.

Authors’ response: The document has been reviewed and the introduction is beefed-up. 

3. Please mention the Pos hoc test performed along with ANOVA in statistical analysis section.

Authors’ response: Pos hoc test performed is now included in S2 Table 

4. Please mention at what point NaHCO3 was added during tablet formulation in methodology section.

Authors’ response: Since direct compression method was used in the preparation of the tablets, NaHCO3 was mixed together with ranitidine HCl, HPMC/NaCMC and MCC first and to this blend magnesium stearate and talc were then added and mixed. 

5. Can author explain the relevance of ex vivo mucoadhesion time as formulation is floating type?

Authors’ response: The formulation is floating, as well as bioadhesive, and swelling type. 

6. What is the rationale to do swelling test till 12 hrs. Discuss in detail regarding effect of polymer and NaHCO3 concentration on swelling and bioadhesion with relevant references of similar work.

Authors’ response: Percent swelling was conducted up to 12 h because the complete swelling was achieved at 12 h. The effects of polymer and NaHCO3 on bioadhesive and swelling characteristics of the formulations is discussed in detail with relevant references in the revised version. 

7. Include standard deviation in table 4.

Authors’ response: Standard deviation is now added to the table. 

8. Elaborate drug release discussion with relevant references.

Authors’ response: Drug release is more elaborated in the revised version. 

9. Author are suggested to cite more recent and relevant references.

Authors’ response: Recent and relevant references are added in the revised version. 

Reviewer #2 

Dear Authors

The present study was to attempt a SR tablet of Ranitidine based on Bio-adhesion mechanism. Study seems to be OK, but not in detailed layout, which was questionable to be published in Plos One. However, results were reported in GOOD state. The final decision will be taken by Eic. I am writing my comments below:

Authors’ response: The mechanism is not only bio-adhesion, rather a combination of floating, bio-adhesion and swelling. 

In Introduction, the actual biopharmaceutical problems of Ranitidine were missing. What is the rationale of preparing its swellable tablet should be mentioned in the Introduction

Authors’ response: The biopharmaceutical problems of ranitidine HCl have been discussed in the manuscript and this was the very reason for designing sustained release through different gastrortentive mechanisms (floating, bioadhesive and swelling). The rationale for swelling as well as for the combination of mechanisms is provided in the introduction. 

In CCD, α-value and design space for the developed formulations? The polynomial equations mentioned by the authors were not in quadratic state, as mentioned in statistical analysis

Authors’ response: The α-value and design space are stated in Table 1. The polynomial equations were quadratic for floating lag time and t50%, and linear for the other response variables (bioadhesion strength, swelling index, drug release after 1 h and drug release after 12 h); consistent in the polynomial and statistical analysis. 

PRESS value seems to be higher in the mathematical models? A justification is needed

What is desirability value of optimized formulation? A graphical optimization graph should be provided?

Authors’ response: The fitness of the model was evaluated by the values of R2. As can be seen in S2 Table, R2 ranged from 0.952 - 0.981 for all selected mathematical model. The ‘Predicted R2’ value was in agreement with the ‘Adjusted R2’ value, the difference being less than 0.2, indicating the reliability of the model. Even though the PRESS values were apparently high in the mathematical models, the selected models have lower PRESS values. The desirability value of the optimized formulation is 0.856 (see line 379, revised manuscript) and the graphical optimization is now included in the revised manuscript (Fig 3). 

Major characterization studies have been done by the Authors, however a factual discussion is missing

Authors’ response: The results that require elaboration are further discussed (effect of the concentration of polymer and effervescent agent on the release and gastro-retentive properties of the matrix tablets is discussed in more detail). Further discussion is also added in the 

mathematical models and statistical analysis in the CCD. Details of the optimized formulation (desirability value, overlay plot) are also incorporated in the revised version. 

What is the rationale of doing the release profile of preliminary batches of CCD?

Authors’ response: The rationale of preliminary studies in CCD were to define the design space and to identify the critical formulation variables. 

What is the floatation behavior of optimized formulation? 

Authors’ response: The floating, bioadhesive, swelling and release characteristics of the optimized formulation were already included in the manuscript and summarized in S2 Fig. 

Overall, the study is not novel, but the presentation is OK

Authors’ response: To our knowledge, there is no report on sustained release ranitidine HCl matrix tablets using a combination of three gastroretentive mechanisms, namely, floating, bio-adhesion and swelling. Hence, we believe this work will contribute significantly to the improvement of drug delivery.

---

## [Editor Report · Decision Letter 1]

4 Jun 2021

Design, Development and Optimization of Sustained Release Floating, Bioadhesive and Swellable Matrix Tablet of Ranitidine Hydrochloride

PONE-D-20-37614R1

Dear Dr. Gebre-Mariam,

We’re pleased to inform you that your manuscript has been judged scientifically suitable for publication and will be formally accepted for publication once it meets all outstanding technical requirements.

Kind regards,

Vineet Kumar Rai, PhD

Academic Editor

PLOS ONE
---

## [Editor Report · Acceptance letter]

18 Jun 2021

PONE-D-20-37614R1 

Design, Development and Optimization of Sustained Release Floating, Bioadhesive and Swellable Matrix Tablet of Ranitidine Hydrochloride 

Dear Dr. Gebre-Mariam:

I'm pleased to inform you that your manuscript has been deemed suitable for publication in PLOS ONE. Congratulations! Your manuscript is now with our production department. 

Kind regards, 

on behalf of

Dr. Vineet Kumar Rai 

Academic Editor

PLOS ONE